# Leveraging Clinical Decision Support and Integrated Medical-Dental Electronic Health Records to Implementing Precision in Oral Cancer Risk Assessment and Preventive Intervention

**DOI:** 10.3390/jpm11090832

**Published:** 2021-08-25

**Authors:** Donald B. Rindal, Patricia L. Mabry

**Affiliations:** HealthPartners Institute, Minneapolis, MN 55425, USA; patricia.l.mabry@healthpartners.com

**Keywords:** oral cancer risk, oropharyngeal cancer risk, precision medicine, screening, SBIRT, prevention, clinical practice, artificial intelligence in health care, implementation

## Abstract

Introduction: Precision medicine is focused on serving the unique needs of individuals. Oral and oropharyngeal cancer risk assessment identifies individual risk factors while providing support to reduce risk. The objective is to examine potential current and future strategies to broadly implement evidence-based oral and oropharyngeal cancer risk assessment and screening in dental practices throughout the United States. Methods: Feasible and effective oral cancer risk assessment and risk reduction strategies, ripe for implementation in dental practice, were identified in the published literature. Results: The Screening, Brief Intervention, Referral for Treatment (SBIRT) model is a feasible approach to assessing individual oral cancer risk and providing risk reducing interventions in the dental setting. HPV is a more recently identified risk factor that dentistry is well positioned to address. Evidence supporting the utilization of specific risk assessment tools and risk reduction strategies is summarized and future opportunities discussed. Discussion: Current knowledge of risk factors for oral and oropharyngeal cancers support the recommendation for dental providers to routinely assess all patients for risk factors, educate them about their personal level of cancer risk, and recommend actions to reduce relevant risk factors. Individuals ages 9–26 should be asked about their HPV vaccination status, educated about HPV and oropharyngeal cancer and receive a recommendation to get the HPV vaccination.

## 1. Introduction

According to the National Institute of Health’s All of Us Research Program [1], precision medicine is focused on serving the unique needs of individuals, taking into account their particular environment, lifestyle, family health history, and genetic makeup. To serve patients in this customized fashion, health care providers need to make customized recommendations for people of different backgrounds, ages, and regions. Personalized information about how to be healthier helps the patient by providing clear guidance on what they can do, and it reduces health care costs by matching the right person with the right treatment the first time. Implementing individual level oral and oropharyngeal cancer risk assessment and preventive interventions in clinical dental practice could be a key strategy for personalizing oral health care.

Each year, an estimated 53,000 US adults are newly diagnosed with oral or oropharyngeal cancer [2]. Visual and tactile examination to detect oral cancers at early stages has not demonstrated improved mortality in asymptomatic people seeking dental care [3,4]. Early diagnosis of potentially malignant lesions holds the potential to improve mortality but current scientific evidence for various proposed techniques is insufficient to support broad clinical adoption [5]. Oral lichen planus lesions have the potential to become malignant [6]. Promising treatments such as topical application of Tacrolimus hold the potential reduce malignant transformation. Inflammatory biomarkers also hold the potential to identify individuals at greater risk of developing oral cancer. One such example is alterations in DNA methylation patterns of genes involved in cell regulation are known to contribute to cancer development [7].

A more upstream approach that identifies and addresses known risk factors offers the potential to reduce oral and oropharyngeal cancer [8]. This primary prevention approach focused on risk factor screening and reduction provides the potential to significantly reduce morbidity and mortality. Oral and oropharyngeal cancers have well-established risk factors [9] of heavy smoking, chewing tobacco, alcohol, marijuana use, chronic inflammation, a weakened immune system, poor oral hygiene, and sexually transmitted high-risk human papillomavirus [10,11,12]. Oral human papillomavirus (HPV) infection is the principal underlying cause of a dramatic increase in oropharyngeal cancer incidence over the last several decades in the United States [13,14,15]. With HPV 16 infection conferring an approximate 20-fold increase in risk of developing oropharyngeal cancer.

Oral cancer risk assessment tools are currently available with electronic versions available in dental schools [16]. The utilization of these tools in a systematic risk screening approach is likely underutilized limiting our ability to prevent oral and oropharyngeal cancers. A National Dental Practice-Based Research Network (National Dental PBRN) study examined health risk assessments conducted by dental practitioners. This study found rates of tobacco screening and discussion (38%) with alcohol screening and discussion (28%) [17].

Incorporating strategies for addressing smoking-cessation with their patients into the curriculum for dental providers (dentists and dental hygienists) is expected to facilitate uptake of smoking cessation counseling into routine dental practice because dental providers initially model their care practices on what they were taught in school [18,19,20]. Health care providers have access to evidence-based guidelines that help patients quit smoking [21,22]. Translation of that knowledge and awareness into daily practice, however, remains low [18,19,20,21,22,23,24,25].

Prevention is defined as the protection of health [26]. Prevention is achieved by identifying its causes and implementing cancer prevention interventions [27]. Our objective is to examine potential current and future strategies to broadly implement evidence-based oral and oropharyngeal cancer risk assessment and screening in dental practices throughout the United States.

## 2. Materials and Methods

This objective is achieved by examining the current peer-reviewed literature to determine what approaches are feasible in the clinical setting and the current evidence on the efficacy of various prevention strategies to address the currently known risk factors. Future opportunities to improve both the accuracy and implementation of risk assessment are also examined in the context of addressing known barriers. These barriers are explored from the perspective of the division of medicine and dentistry and the changing trends in dental practice. These trends include the integration of dental care delivery into health care systems such as Federally Qualified Health Centers, HealthPartners in Minnesota, Kaiser Permanente in Oregon, and the Marshfield Clinic in Wisconsin. Dental practice models are also changing with an increase in large dental service organizations with more collective resources to support a more robust oral cancer risk assessment program. The two authors conducted the literature searches identifying current evidence and the research gaps.

## 3. Results

### 3.1. Current Opportunities to Implement Precision in Oral Cancer Risk Assessment and Preventive Intervention in Practice

#### 3.1.1. Screening, Brief Intervention, Referral for Treatment (SBIRT)

Screening, brief intervention and referral to treatment (SBIRT) is an evidenced-based practice that that encourages healthy behaviors, identifies problematic drug and alcohol use early, reduces substance misuse, and refers individuals to treatment if they need it. It involves a process of steps that a healthcare professional including dental clinicians can take to assess alcohol and drug use behaviors in their patients to reduce oral cancer risk along with other risks to their health [28]. This approach is feasible in clinical settings and aligns well with a personalized approach.

Current use of SBIRT in dental settings is limited [29,30]. A 2013 survey of dentists participating in a dental practice-based research network [31] found that few respondents provided any follow-up interventions or referrals to treatment even when they screened for substance use. Counseling or referrals following positive tobacco use screenings were more than twice as common as for positive alcohol or drug use screenings. Several studies have tested the delivery of brief behavioral interventions focused on tobacco cessation and found them to be effective [32,33,34,35]. A recent international Cochrane review of trials in dentistry (including 13 U.S. studies) found overall positive (but low-certainty) evidence that brief intervention from dental providers alone increases quit rates, although the level of evidence was more robust (moderate-certainty) when such brief intervention was paired with access to pharmacotherapy [36]. Research on SBIRT for alcohol and other substance use in dentistry is far more limited. Neff and colleagues [37,38] conducted a NIDCR-funded cluster-randomized trial of screening and brief intervention in 13 dental practices for 103 patients reporting heavy drinking. Compared to participants who received usual care in control clinics, participants who received a brief intervention reported decreased quantity and frequency of alcohol consumption at 6-month follow-up. Thus, there is evidence that brief interventions for tobacco and alcohol can be effective in the dental setting. More research is needed to establish the effectiveness of SBIRT services for addressing other substance use in dental practice. Moreover, there is a need to identify approaches to SBIRT that leverage the growing integration of dentistry within larger health systems and Federally Qualified Health Centers.

Research on barriers to dental providers addressing substance use is limited. However, over a decade ago Neff and colleagues examined factors affecting readiness to screen or provide brief intervention for substance use in dental practice [39]. Perceived effectiveness/efficiency concerns, the perceived appropriateness of addressing substance use in dental settings, the dental professional’s role, the need for training, and concern about time and reimbursement were among the barriers noted at the time. Similarly, McNeely’s 2013 survey [31] identified a lack of knowledge or training as perceived barriers among dental providers. While much has been learned regarding the implementation of SBIRT in general healthcare settings over the past decade, lack of knowledge or training may continue to be barriers to addressing substance use in dental practice.

#### 3.1.2. HPV Vaccination Recommendation

Persistent HPV infection is a well-established risk factor for developing oropharyngeal cancer [40]. Most dental providers understand that HPV is a sexually transmitted infection and know an HPV vaccine is available, but many are not discussing the oropharyngeal cancer link or recommending vaccination [41]. They are less often to recommend HPV vaccination if they were uncomfortable discussing sex, perceive hesitancy in the parents or perceive the patient is low risk. In addition, many providers believe this is not their responsibility [41].

Uptake of the HPV vaccine remains modest, despite evidence that vaccine-type HPV prevalence is decreasing as a result of HPV vaccination [42]. Dental providers participation in the effort to increase HPV vaccination uptake could help fill this gap [43]. The National HPV Vaccination Roundtable has developed an action guide specific to dental providers [9].

### 3.2. Future Opportunities

#### 3.2.1. Integration of Dentistry with Medical Practice

Opportunities for collaborative practice between dentists and physicians can occur in health care systems with a dental component. The full integration of dentistry in health care teams remains unrealized leaving dentistry on the margins of new innovative new models of care delivery [44]. The need to integrate medical and dental patient care is clear; yet doing so in practice faces several challenges. Compatible and interoperable EHR systems are needed to establish formal communication, collaboration, and referral networks across these professions at all levels [45]. Moreover, few organizations are currently positioned to address this issue as medical and dental practices continue to remain siloed in their health-care delivery [46]. In order to seamlessly share patient information across medical and dental providers, health informatics and information technology can lead to solutions that facilitate collaboration and communication between health-care providers [47]. Linking dental providers with pediatricians, primary care providers and behavioral health is essential to ensuring the continuum of care where patients get the appropriate additional services.

Systems of care in which teams are currently practicing integrated oral health care delivery with an integrated EHR include Permanente Dental Associates and HealthPartners [48]. These organizations are early adopters who envision the benefits while identifying the challenges to be addressed in optimizing the potential benefits for both providers and patients. A survey study of dentists and physicians examined the value of sharing EHR information [49]. Interoperable EHRs could facilitate information transfer between providers and enhance research on oral–systemic health connections. Both dentists and physicians believe an interoperable EHR would be useful to practice, but information needs are different between these groups leading to the conclusion that refinement of information will optimize the benefits.

We are seeing some progress in EHR software companies such as Epic [50] integrating oral health into their record systems [51,52,53]. Currently this integration is occurring in community health centers [54], universities, and larger dental service organizations along with health care systems that include dentistry. New opportunities may develop as dental practice setting trends are shifting from private practices to dental service organizations [55].

#### 3.2.2. Clinical Decision Support

Electronic health record-based clinical decision support (CDS) “provides clinicians, staff, patients, or other individuals with knowledge and person-specific information, intelligently filtered or presented at appropriate times, to enhance health and health care” [56]. CDS can improve clinical care in various health care settings when implemented in conjunction with additional care improvement strategies [57]. Recent advances in health information technology have yielded CDS tools embedded in electronic health records (EHRs) that can provide information to assist healthcare providers or display suggestions for the clinician or patient to consider, helping to overcome some of the knowledge, training, and role concern barriers identified by previous dental SBIRT researchers. CDS that reminds and helps providers deliver smoking cessation interventions by providing evidence-based information during care delivery in a clinically relevant format holds the potential to facilitate an evidence-based practice approach [58]. Coupled with growing inclusion of dentistry within larger health systems and the integration of dental and health records under a common electronic health record (EHR), these advances in CDS could be transformative in individualizing care for patients, including for oral cancer risk assessment and preventive intervention.

A tobacco CDS tool increased dental practitioners’ advice to quit smoking and referral to a quitline during a group randomized trial. After completion of the trial, the CDS was implemented in all the dental clinics within HealthPartners. Sustained use was examined, and the results showed that the dental practitioners persisted in using the CDS tool to refer smokers to a quitline [59]. This ongoing utilization is more likely to occur if the CDS tool is designed with significant involvement of key stakeholders [60].

In summary, an integrated EHR enhances the information needed to make better diagnoses and treatment decisions. These benefits need to be balanced with extra time spent in front of a computer searching for relevant information and entering information about the visit [52]. CDS can address some of these challenges by summarizing relevant information into one interface that supports better decision making and messaging to patients while simplifying documentation of the visit [32].

#### 3.2.3. Quality Measures and Payment

The dental profession is experiencing disruptions in technology, communications, workforce, payment, and management, all driven by creative innovators sponsored by venture capital, nonprofits, and governments [61]. Two disruptions with the potential to significantly advance implementation of a personalized risk reduction approach to oral/oropharyngeal cancer are quality measures and value-based care.

Quality measures are not well developed in dentistry. A recent systematic review examined existing quality measures in the field of oral health care. This review identified measures of quality pertaining to treatment and preventive services (*n* = 71). Development of these measures often lacked involvement of patients and dental professionals. Few projects reported on the validity (*n* = 2) and reliability (*n* = 3) of the measures, and few (*n* = 3) piloted the measures in practice [62]. None of these measures are widely implemented and reported. Oral/oropharyngeal cancer prevention could be advanced by broad adoption of quality measures related to addressing tobacco use, alcohol/substance use, and HPV vaccination status.

Value-based healthcare is a healthcare delivery model in which providers are paid based on patient health outcomes and achieving quality metrics. Providers are rewarded for helping patients improve their health, reduce the effects and incidence of disease, and live healthier lives in an evidence-based way. Fee-for-service pays providers based on the amount of healthcare services they deliver. The “value” in value-based healthcare is derived from measuring health outcomes against the cost of delivering the outcomes [63]. Value payments tied to quality measures could facilitate adoption of risk assessment and prevention. A recent publication proposed a tentative set of seven value measures. The measures comprise the four dimensions of oral health–related quality of life (oral function, orofacial pain, appearance, and psychosocial impact) and the three dental conditions with the highest burden of disease. (untreated caries, untreated periodontal pockets, and tooth loss) [64].

#### 3.2.4. Artificial Intelligence

Artificial Intelligence (AI) refers to the ability of machines (i.e., computers) to perform tasks normally performed by humans. The potential for Artificial Intelligence (AI) to transform healthcare delivery is tremendous [65]. The main reason for this is the ability of AI to identify complex patterns in vast amounts of data. Consider that deep learning algorithms (a form of AI) have excelled at the task of image classification to the point that, without human supervision, computers have learned to distinguish between images of cats vs. dogs (and even to distinguish between dog breeds) [66] and, in spectacular fashion, to enable self-driving cars [67]. With respect to precision oral health, AI could potentially be used to help identify which treatments or combinations are best suited for a particular individual based on a large number of factors, such as age at diagnosis, comorbid conditions, current and past medication use, and lifestyle factors. Perspectives on the potential for AI in dentistry are presented in greater depth elsewhere [66,68,69,70].

To date, AI in health care has had the greatest utility in the area of aiding diagnostic imaging but has seen little uptake to date due to lack of data which is due to privacy concerns and logistical issues in accessing the data. Several issues have prevented AI from enjoying widespread adoption, despite its promise. Lack of a financial reimbursement model for AI diagnosis is one obvious barrier discussed in detail elsewhere [71]. Another significant issue is that to conduct AI/ML one needs large amounts of data, and for AI applications in medicine/dentistry this has to be patient data, which means accessing it requires dealing with privacy, security, consent, and other issues pertaining to sensitive data. Even if these barriers were overcome there is a dearth of EHR data that is in ready-to-use format for AI/ML. This is because health care systems keep their EHR data in different formats and ontologies and not easily combined. Often the data would need to be converted to another data format before it can be used for computationally intense analyses such as AI/ML. Other investments include cloud compute/storage and access to Graphical Processing Units. Another significant barrier to AI adoption is trust. Early efforts to incorporate AI/ML in health care resulted in some now-famous cases that raised alarm for perpetuation of racial bias in health care delivery [72]. On the positive side this led to widespread awareness that AI/ML can be inherently dangerous in this regard and helped to promote further attention to methods and considerations to ensure fairness and trust. On the negative side, it fed a fear that AI/ML could create inequality [73]. A tendency for slow adoption may be especially true when the new methodology, AI/ML, is not well understood. AI/ML has been widely criticized for being a “black box” [74] meaning that the algorithms take data in and churn out predictions, but how they make those predictions is not well understood. There is now a growing call for “explainable AI” to address this problem [75]. Any AI application in dentistry should demonstrate tangible value by improving access, quality of care, increasing efficiency and safety, empowering patients, supporting research, or increasing sustainability [68].

Considering that NIH is making investments to pave the way for AI in health research [76], and that even bigger investments are expected in the near future [77], dental practitioners can expect that AI will start to transform dental practice in the not-too-distant future. As described in an NIDCR news item [78], AI has already been able to detect the presence of “molecular and genetic alterations based only on tumor images across 14 cancer types, including those of the head and neck” [79]. In anticipation of AI’s paradigm shifting role in health care, training programs directed at health care professionals have recently launched at MIT [80], Stanford [81], and elsewhere [82], and it seems similar programs directed at dental professionals are sure to follow. This section may be divided by subheadings. It should provide a concise and precise description of the experimental results, their interpretation, as well as the experimental conclusions that can be drawn.

## 4. Discussion

Current knowledge of risk factors for oral and oropharyngeal cancers support the need to routinely assess all patients for risk factors, educating patients about cancer risk and recommending actions to reduce the relevant risk factors. Implementing SBIRT into all dental practice settings is a feasible and effective approach to address substance use with patients while addressing perceived and real barriers. Individuals ages 9–26 should be asked about their HPV vaccination status. Dental providers should educate patients and parents of age-eligible children about the link between HPV and oropharyngeal cancers which are impacting younger adults and recommend HPV vaccination as cancer prevention.

For patients using tobacco products, unsafe alcohol use, and other high-risk substances the dental provider should educate the patient about the cancer risks, provide a brief intervention supporting the patient on quitting and provide a referral to a tobacco quit line or a substance use specialist for further assessment and support. In healthcare systems that include dental utilizing an integrated EHR these activities can be further supported by utilizing CDS tools that systematically support the dental provider in collecting the risk information, prompting further assessment questions and providing recommended scripts based on motivational interviewing. For dental providers embedded within health care systems this can be further augmented by facilitating the referral of the patient to other medical services addressing substance use or initiating the vaccine series.

The American Dental Association has taken leadership in developing quality measures for dentistry. The Dental Quality Alliance has published measures that will be used by Medicaid and dental plans to measure quality [83]. In addition, the Wisconsin Collaborative for Healthcare Quality [84], formed to develop performance measures that improves the quality and affordability of health care in Wisconsin, has included dental measures to be developed. Their members include members includes 35 health systems, 325 medical clinics, and more than 150 dentists. Collectively, these efforts hold the potential to move quality measures into the forefront of dentistry with the desired result of increasing evidence-based care. Adding measures related to tobacco and substance use along with HPV vaccination could significantly advance implementation of risk prevention measures in dentistry. When payment plans align with these measures, broad implementation is more likely to occur.

Value-based payment (VBP) arrangements are growing in health care but just emerging in oral health care. Embarking on a new payment arrangement presents new challenges. Probably the biggest challenge is related to the electronic data currently available at the practice level. Significant gaps exist in coding, data collection, exchange, and analysis [85]. VBP can provide financial resources to fill critical infrastructure gaps such as electronic health records, decision support, and care management tools. Combining quality measures and VBP holds good potential advance the implementation of oral/oropharyngeal cancer risk assessment and risk reduction into daily practice where it can benefit patients.

When AI makes inroads into dentistry, we see opportunities to incorporate AI findings into the CDS platform. The future advances will happen when we have better EHR data and systems that can integrate CDS. The time is not now but we need to start planning for that future by thinking about what can be done now to prepare. In addition, research is examining the relationship of the oral microbiome and various cancers including oral cancer [86]. If new risk factors are identified that more precisely predict oral or oropharyngeal cancer risk, the CDS could incorporate new evidence brought to the dental provider at the point of care.

Advancing the implementation oral and oropharyngeal cancer risk assessment in dentistry will start with larger group practices who have invested in better EHR systems and are linked with academics and researchers with a care agenda prioritizing risk assessment and risk reduction. These leaders will also see the value in better data, quality measures, and value-based care. These early adopters will need to demonstrate better health outcomes, reducing costs, and improving the patient experience which aligns with the triple aim as defined by the Institute for Healthcare Improvement [87]. Once the profession reaches that critical mass, risk assessment will be a routine part of daily clinical practice.

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
