# Peer review of "Leveraging Clinical Decision Support and Integrated Medical-Dental Electronic Health Records to Implementing Precision in Oral Cancer Risk Assessment and Preventive Intervention"

_jpm, 2021, doi:10.3390/jpm11090832_

Round 1
Reviewer 1 Report
In the manuscript entitled: “Leveraging Clinical Decision Support and Integrated Medical- Dental Electronic Health Records to Implementing Precision in Oral Cancer Risk Assessment and Preventive Intervention”, the authors evaluated potential current and future strategies to broadly implement evidence-based oral and oropharyngeal cancer risk assessment and screening in dental practices throughout the United States.
The authors found that the Screening, Brief Intervention, Referral for Treatment (SBIRT) model is a feasible approach to assessing individual oral cancer risk and providing risk reducing interventions in the dental setting. HPV is a more recently identified risk factor that dentistry is well positioned to address.
The authors concluded that current knowledge of risk factors for oral and oropharyngeal cancers support the recommendation for dental providers to routinely assess all patients for risk factors, educate them about their personal level of cancer risk, and recommend actions to reduce relevant risk factors. Individuals ages 9-26 should be asked about their HPV vaccination status, educated about HPV and oropharyngeal cancer and receive a recommendation to get the HPV vaccination.
Major comments:
In general, the idea and innovation of this study, regards analysis of Precision in 3 Oral Cancer Risk Assessmentis interesting, because the role of these factors in dentistry are validated but further studies on this topic could be an innovative issue in this field could be open a creative matter of debate in literature by adding new information. Moreover, there are few reports in the literature that studied this interesting topic with this kind of study design.
The study was well conducted by the authors; However, there are some concerns to revise that are described below.
The introduction section resumes the existing knowledge regarding the important factor linked with oral cancer and oral preneoplastic disorders.
However, as the importance of the topic, the reviewer strongly recommends, before a further re-evaluation of the manuscript, to update the literature through read, discuss and must cites in the references with great attention all of those recent interesting articles, that helps the authors to better introduce and discuss the role of related inflammatory biomarkers and therapy of oral lichen planus and OSCC: 1) Polizzi A, Santonocito S, Lo Giudice A, Alibrandi A, De Pasquale R, Isola G. Analysis of the response to two pharmacological protocols in patients with oral lichen planus: A randomized clinical trial. Oral Dis. 2021 Jul 12. doi: 10.1111/odi.13960. 2) Ferlazzo N, Currò M, Zinellu A, Caccamo D, Isola G, Ventura V, Carru C, Matarese G, Ientile R. Influence of MTHFR Genetic Background on p16 and MGMT Methylation in Oral Squamous Cell Cancer. Int J Mol Sci. 2017 Mar 29;18(4):724. doi: 10.3390/ijms18040724.
In the material and methods section, should better clarify the integration of dentistry with medical practice. Moreover, please more specify the scientists involved in the different stages of the study.
The discussion section appears well organized with the relevant paper that support the conclusions, even if the authors should better discuss the relationship between OPMD therapy and OSCC development. The conclusion should reinforce in light of the discussions.
In conclusion, I am sure that the authors are fine clinicians who achieve very nice results with their adopted protocol. However, this study, in my view does not in its current form satisfy a very high scientific requirement for publication in this journal and requests a revision before a futher re-evaluation of the manuscript.
Minor Comments:
Abstract:
- Better formulate the abstract section by better describing the aim of the study
Introduction:
- Please refer to major comments
Discussion
- Please add a specific sentence that clarifies the results obtained in the first part of the discussion
Page 5 last paragraph: Please reorganize this paragraph that is not clear
Author Response
Reviewer 1
Comments and Suggestions for Authors
In the manuscript entitled: “Leveraging Clinical Decision Support and Integrated Medical- Dental Electronic Health Records to Implementing Precision in Oral Cancer Risk Assessment and Preventive Intervention”, the authors evaluated potential current and future strategies to broadly implement evidence-based oral and oropharyngeal cancer risk assessment and screening in dental practices throughout the United States.
The authors found that the Screening, Brief Intervention, Referral for Treatment (SBIRT) model is a feasible approach to assessing individual oral cancer risk and providing risk reducing interventions in the dental setting. HPV is a more recently identified risk factor that dentistry is well positioned to address.
The authors concluded that current knowledge of risk factors for oral and oropharyngeal cancers support the recommendation for dental providers to routinely assess all patients for risk factors, educate them about their personal level of cancer risk, and recommend actions to reduce relevant risk factors. Individuals ages 9-26 should be asked about their HPV vaccination status, educated about HPV and oropharyngeal cancer and receive a recommendation to get the HPV vaccination.
Major comments:
In general, the idea and innovation of this study, regards analysis of Precision in 3 Oral Cancer Risk Assessments interesting, because the role of these factors in dentistry are validated but further studies on this topic could be an innovative issue in this field could be open a creative matter of debate in literature by adding new information. Moreover, there are few reports in the literature that studied this interesting topic with this kind of study design.
The study was well conducted by the authors; however, there are some concerns to revise that are described below.
Response: Thank you for the positive feedback and for the suggestions to improve the manuscript.
The introduction section resumes the existing knowledge regarding the important factor linked with oral cancer and oral preneoplastic disorders.
However, as the importance of the topic, the reviewer strongly recommends, before a further re-evaluation of the manuscript, to update the literature through read, discuss and must cites in the references with great attention all of those recent interesting articles, that helps the authors to better introduce and discuss the role of related inflammatory biomarkers and therapy of oral lichen planus and OSCC: 1) Polizzi A, Santonocito S, Lo Giudice A, Alibrandi A, De Pasquale R, Isola G. Analysis of the response to two pharmacological protocols in patients with oral lichen planus: A randomized clinical trial. Oral Dis. 2021 Jul 12. doi: 10.1111/odi.13960. 2) Ferlazzo N, Currò M, Zinellu A, Caccamo D, Isola G, Ventura V, Carru C, Matarese G, Ientile R. Influence of MTHFR Genetic Background on p16 and MGMT Methylation in Oral Squamous Cell Cancer. Int J Mol Sci. 2017 Mar 29;18(4):724. doi: 10.3390/ijms18040724.
Response: We have added a paragraph to the introduction that incorporates these references. Our approach was to make the clinician reader aware of these promising future approaches while making the case for primary prevention. We anticipate that other contributors to this special issue will provide a more extensive review of these approaches based on our understanding of the focus of this special issue.
In the material and methods section, should better clarify the integration of dentistry with medical practice. Moreover, please more specify the scientists involved in the different stages of the study.
Response: We have added text related to the integration of dentistry with medicine and text regarding the roles of the scientists.
The discussion section appears well organized with the relevant paper that support the conclusions, even if the authors should better discuss the relationship between OPMD therapy and OSCC development. The conclusion should reinforce in light of the discussions.
Response: The authors have decided that discussing OPMD therapy would not fit well with the current discussion. We thought it was valuable to include this in the introduction, but our primary objective was to identify actions by dental providers that should be implemented now in clinical practice. This objective aligns with our discussion with the guest editor when exploring the value of our potential contribution.
In conclusion, I am sure that the authors are fine clinicians who achieve very nice results with their adopted protocol. However, this study, in my view does not in its current form satisfy a very high scientific requirement for publication in this journal and requests a revision before a further re-evaluation of the manuscript.
Response: Both authors are researchers and one author practices one day per week. In discussing a possible manuscript with the guest editor, it was determined that a clinician focused article examining current evidence-based approaches and potential future approaches would be of value for this special edition. Therefore, the objective of this manuscript is to bring forth evidence-based recommendations for implementing better oral cancer care in daily clinical practice.
Minor Comments:
Abstract:
- Better formulate the abstract section by better describing the aim of the study
Response: We have edited the abstract (see track changes)
Introduction:
- Please refer to major comments
Response: See our responses and edits for that section.
Discussion
- Please add a specific sentence that clarifies the results obtained in the first part of the discussion
Response: A sentence has been added that highlights what can be done now in all settings.
Page 5 last paragraph: Please reorganize this paragraph that is not clear
Response: The last paragraph of page 5 now resides on page 6. It is the third paragraph of the discussion. We have reorganized to provide greater clarity.
Reviewer 2 Report
The paper only appears to touch on the topic and not make any sound suggestions nor give any relevant examples.
For example, with regard to using EHR - there is no indication with regard to how easy these are to use or to integrate into existing systems.
With regard to the CDS, more examples of which centres have managed to incorporate this is required, plus a discussion of how successful the incorporation has been.
The value-based healthcare is an interesting concept - but no discussion on how "value" can be measured, nor examples of where and how this has been used.
With regard to A.I - a much, much deeper discussion of why this has not been fully incorporated into the current healthcare system is required!
Author Response
Reviewer 2
The paper only appears to touch on the topic and not make any sound suggestions nor give any relevant examples. For example, with regard to using EHR - there is no indication with regard to how easy these are to use or to integrate into existing systems.
Response: We have added a paragraph on page 4 in the section on integration. The text added was based on our search of the peer reviewed literature on provider experience with these systems. We also made other significant additions to add more depth to the discussion of the topic. We trust that these collective additions address the overarching concern.
Regarding the CDS, more examples of which centers have managed to incorporate this is required, plus a discussion of how successful the incorporation has been.
Response: We added a paragraph providing an example of a CDS that was broadly implemented and sustained use was documented with data. We provide references to support this example and some explanation as to why it was used.
The value-based healthcare is an interesting concept - but no discussion on how "value" can be measured, nor examples of where and how this has been used.
Response: We have added additional text that includes proposed measures. We found several identify value-based payment examples in medicine but none in dentistry. All the references we found were conferences and papers addressing what it is and how dentistry might move toward value-based care. That is why we discuss it from the perspective of potential future opportunities.
Regarding A.I - a much, much deeper discussion of why this has not been fully incorporated into the current healthcare system is required!
Response: We have added significant text and references to expand on the reasons. This includes what will need to be addressed in dentistry before there is broad adoption.
Thank you for the excellent suggestions.
Round 2
Reviewer 1 Report
The authors have well addressed to all comments raised by the reviewer. There are no further issues.